# Epidemiological models for predicting Ross River virus in Australia: A systematic review

Wei Qian[1], Elvina Viennet[2,3], Kathryn Glass[4], David Harley[1]*

**1** Mater Research Institute-University of Queensland (MRI-UQ), Brisbane, Queensland, Australia,
**2** Research and Development, Australian Red Cross Lifeblood, Brisbane, Queensland, Australia, **3** Institute for Health and Biomedical Innovation, School of Biomedical Sciences, Queensland University of Technology (QUT), Queensland, Australia, **4** Research School of Population Health, Australian National University, Acton, Australian Capital Territory, Australia

☯ These authors contributed equally to this work.
* d.harley@uq.edu.au

**Data Availability Statement:** All relevant data are within the manuscript and its Supporting Information files.

## Abstract

Ross River virus (RRV) is the most common and widespread arbovirus in Australia. Epidemiological models of RRV increase understanding of RRV transmission and help provide early warning of outbreaks to reduce incidence. However, RRV predictive models have not been systematically reviewed, analysed, and compared. The hypothesis of this systematic review was that summarising the epidemiological models applied to predict RRV disease and analysing model performance could elucidate drivers of RRV incidence and transmission patterns. We performed a systematic literature search in PubMed, EMBASE, Web of Science, Cochrane Library, and Scopus for studies of RRV using population-based data, incorporating at least one epidemiological model and analysing the association between exposures and RRV disease. Forty-three articles, all of high or medium quality, were included. Twenty-two (51.2%) used generalised linear models and 11 (25.6%) used time-series models. Climate and weather data were used in 27 (62.8%) and mosquito abundance or related data were used in 14 (32.6%) articles as model covariates. A total of 140 models were included across the articles. Rainfall (69 models, 49.3%), temperature (66, 47.1%) and tide height (45, 32.1%) were the three most commonly used exposures. Ten (23.3%) studies published data related to model performance. This review summarises current knowledge of RRV modelling and reveals a research gap in comparing predictive methods. To improve predictive accuracy, new methods for forecasting, such as non-linear mixed models and machine learning approaches, warrant investigation.

## Author summary

As the most common human arbovirus infection in Australia, Ross River virus exerts a significant public health and economic burden on the population. Because the virus is transmitted by mosquitoes, incidence is influenced by climate, environment, and socio-economic factors. Using epidemiological models to predict incidence or outbreaks of RRV fully utilises these data to inform decision-making. In this systematic review, we

**Funding:** This work is supported by the University of Queensland Research Training Scholarship and Frank Clair Scholarship. The funders had no role in study design, data collection and analysis, decision to publish, or preparation of the manuscript.

**Competing interests:** The authors have declared that no competing interests exist.

summarised models and their predictive performance, and highlighted significant exposures in order to increase understanding of transmission.

## Introduction

Ross River virus, a mosquito-transmitted *Alphavirus*, is the most common arboviral infection of humans in Australia [1, 2] and often results in a characteristic syndrome, including constitutional effects, rash, and rheumatic manifestations [1, 3]. A total of 123,875 cases of RRV infection were reported from 1993 to 2019 in Australia, of which nearly half (48.8%) were from Queensland. [4]

Ross River virus transmission is primarily influenced by mosquito abundance, reservoir host populations, and climatic, environmental (e.g. rainfall, temperature, tides, river flow, vegetation cover) and socio-economic factors (e.g. urban development, housing infrastructure) [1, 5–9]. Models of RRV using these exposures can improve knowledge of RRV transmission or be used to give early warning of outbreaks, thus aiding disease prevention and control. However, the relationships between exposures and RRV incidence are complex. For instance, climate can influence vector abundance, host populations and the behaviour of vectors and hosts, and climate and weather are influenced by human behaviour (e.g. global warming, heat island effects, effects of large dams) and geographical factors like altitude [10, 11]. Therefore, exposures do not have a simple correlation with disease incidence, which increases the difficulty of forecasting.

Generalised linear regression and time-series models are widely used for infectious disease prediction [12]. Linear regression models are straightforward, but often inadequate for prediction in complex systems. Time-series models are especially suitable for analysing data containing autocorrelation and which shows periodic fluctuations [13, 14].

Three reviews on exposures or predictive models of RRV have been published, however all concentrated on a description of exposures and their relationships with disease, with less attention to models and their performance, and none were systematic reviews. A review by Tong et al. (2008) [8] included more than 15 articles on predictors of RRV transmission. Analytical methods were listed, and the detailed research process and results were described to elucidate the association between climatic, social and environmental factors and RRV disease. Yu et al. (2014) [15] identified research on the impact of climate change on RRV disease. All models applied in these studies were listed, but the characteristics of the models were not discussed. Another review by Jacups et al. (2008) [9] described the vectors and vertebrate reservoirs of RRV, the possible impact of climate change on incidence, and summarised the models and the climatic factors applied in 15 studies. RRV models were discussed in this study, but the focus was on the influence of covariates and the geographical size of the study areas on the model accuracies. However, these three reviews neither provide a detailed profile of the models nor quantification of their performance.

In this review, the research hypothesis is that a detailed summary of all available primary modelling research for RRV enables an evaluation of the effectiveness of the models in forecasting disease and improving knowledge of exposures and transmission cycle. We aim to describe modelling approaches applied in RRV disease prediction in Australia, the performance of these models, the variables used and the models' performance in prediction.

## Methods

This systematic review followed the Preferred Reporting Items for Systematic Reviews and Meta-Analyses (PRISMA) guidelines [16]. The PRISMA Checklist is available in S1 Table. The proposal for this study was completed before data extraction (S1 Text).

## Literature search strategy

We performed a structured literature search using PubMed, EMBASE, Web of Science, Cochrane Library, and Scopus for articles published between January 1, 1980 and January 21, 2020 with search terms encompassing pathogen (i.e. "Ross River virus"), methods (i.e. "model" OR"forecast"), and exposures (i.e. "impact factor" OR "predictor" OR "association"). Articles not relevant to our aims were excluded (i.e. "gene" OR "protein" OR "transfusion"). These search terms efficiently excluded irrelevant records. Studies on genes or proteins were mainly laboratory-based and not relevant for epidemiological risk prediction. Studies on transfusion-related RRV transmission were excluded because such transmission is infrequent and can be ignored. In addition, findings could not feasibly be integrated with studies of mosquito-transmitted disease. Search terms are provided in S2 Text.

## Inclusion and exclusion criteria

Studies of RRV using population-based data, incorporating at least one epidemiological model and analysing the association between an exposure or exposures and RRV incidence or outbreaks, located in Australia, in English and with full-text available were included in this review. Review articles, meeting abstracts, letters, books, reports and comments were excluded. Studies on RRV virology, vaccines, and animal models were also excluded. Articles describing models of RRV vectors or non-human reservoir hosts without human epidemiological data were excluded, as were studies involving transmission dynamic modelling. The records were first screened by titles and abstracts, then the full texts were reviewed before a final decision on inclusion. Study inclusion was conducted by one author (WQ), and in cases of uncertainty, all four authors reached a decision after discussion.

For included records, title, author, publication year, research area and period, predictors and format of predicted outcome, modelling method, significant results, prediction performance, model evaluation and model validation were extracted.

## Methodological quality assessment and data extraction

Included studies were assessed according to recently published criteria for observational studies [17–20]. Each criterion was scored 2, 1 or 0 if the studies fully, partly or barely met the criterion (S2 Table). The statement of funding and conflict of interest were each scored 1 if they were stated clearly. The studies were classified into three groups depending on total scores: high (19–24), medium (13–18), and low quality (<13). Related study registrations were searched to evaluate publication bias. Data extraction and quality assessment were conducted by one author (WQ) and discussed by all authors where there were uncertainties.

All the authors participated in the entire review process, discussed the main decisions and reached agreement on study selection, data extraction and study assessment together.

Study characteristics and model performance were tabulated. Exposures applied in these models were summarised and their association with RRV listed.

The transmission cycle of RRV and key exposures influencing RRV infection are illustrated in Fig 1.

## Results

After duplicates were removed from 2,227 searched records, we screened 976 papers; after exclusion criteria were applied, 43 records remained (Fig 2) [21–63]. All studies were published in the last 20 years, and 19 (44.2%) during the past decade. Nearly half the studies were conducted in Queensland (20, 46.5%), while five (11.6%) were in Western Australia.

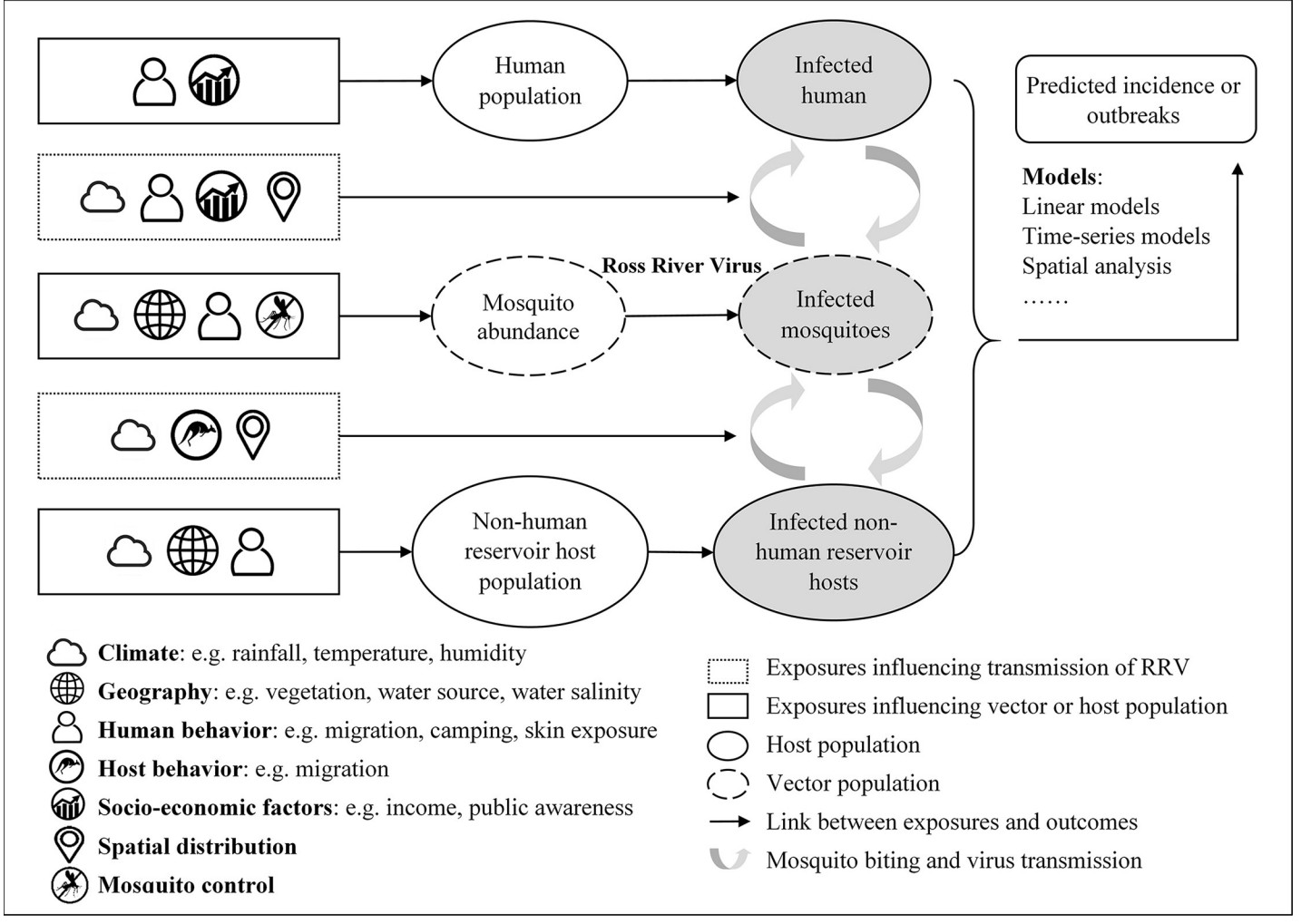

**Fig 1. Key elements of RRV prediction models.**

The quality scores for the studies are listed in Table 1. Detailed scores are provided in S3 Table. All studies attained high (33, 76.7%) or medium quality scores (10, 23.3%). There were two articles published without significant results. No systematic review registration related to RRV modelling was found.

Models were grouped into three categories: generalised linear models (22 studies, 51.2% of 43 studies); time-series models (11, 25.6%); other models including Cumulative Sum based (CUSUM-based) methods (3, 7.0%), spatial or temporal analysis (5, 11.6%), Classification and Regression Tree (CART) (2, 4.7%), Maxent model, Hurdle model, Besag, York, and Mollié (BYM) model and generalised additive model (1 each, 2.3%) (Tables 2–4). In some studies, different types of models, or models with different covariates were compared. A total of 140 models were built in these 43 studies.

Fourteen articles (32.6% of 43 articles) used mosquito abundance or related data as model covariates. Climate and weather data were used by 27 of the 43 studies (62.8%). Other exposures such as river flow, distance to surface water sources, historical RRV cases and host population were also used (Table 5). Rainfall (applied in 69 models, 49.3% of 140 models),

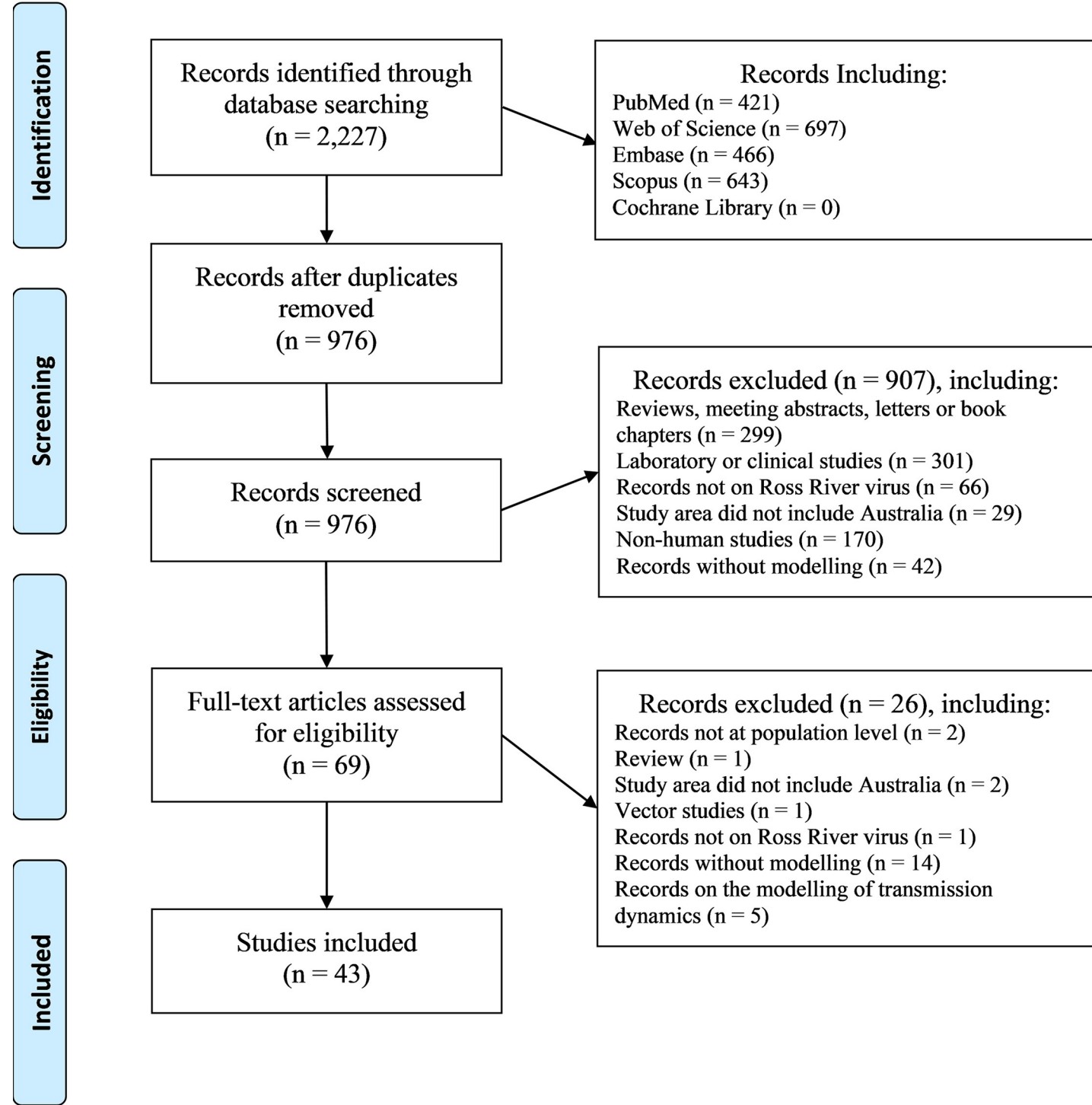

**Fig 2. Flow chart of study selection process.**

temperature (66, 47.1%) and tide height (45, 32.1%) were the three most commonly used risk factors for predicting RRV infection. The full list of 65 exposures in 7 categories, their associations with RRV and time-lags are provided in S4 Table.

**Table 1. Score for quality assessment criteria.**

| Assessment criterion | Number (%) of articles scoring 2* | Number (%) of articles scoring 1 | Number (%) of articles scoring 0 |
|---|---|---|---|
| 1. Clarity of aims and objectives | 43 (100.0) | 0 (0.0) | 0 (0.0) |
| 2. Definition of study geographical area | 40 (93.0) | 3 (7.0) | 0 (0.0) |
| 3. Data source of RRV clearly described | 42 (97.7) | 1 (2.3) | 0 (0.0) |
| 4. Data source of covariates clearly described | 43 (100.0) | 0 (0.0) | 0 (0.0) |
| 5. Data quality considered | 22 (51.2) | 5 (11.6) | 16 (37.2) |
| 6. Model structure clearly described and appropriate for the research question | 41 (95.3) | 2 (4.7) | 0 (0.0) |
| 7. Modelling methods appropriate for the research question | 42 (97.7) | 1 (2.3) | 0 (0.0) |
| 8. Model evaluation | 25 (58.1) | 3 (7.0) | 15 (34.9) |
| 9. Model validation | 14 (32.6) | 1 (2.3) | 28 (65.1) |
| 10. Results clearly and completely presented | 36 (83.7) | 7 (16.3) | 0 (0.0) |
| 11. Results appropriately interpreted and discussed in context | 40 (93.0) | 3 (7.0) | 0 (0.0) |
| 12. Funding statement | - | 34 (79.1) | 9 (20.9) |
| 13. Conflict of interest statement | - | 17 (39.5) | 26 (60.5) |

*The assessment of Funding statement and Conflict of interest statement was scored 1 or 0.

Most studies (23, 53.5% of 43 studies) used incidence rates or disease occurrence as dependent variables (Table 2). Twelve studies used outbreaks as dependent variables (Table 3), while ten used notified cases (Table 4). Two articles used incidence rates and outbreaks in different models. Linear models were applied for analysing all forms of notified data. Most studies using time-series and spatial analysis models forecast incidence rather than outbreaks. All studies using CUSUM-based models predicted outbreaks.

Only ten of the 43 studies published data related to model performance; among them, five used logistic regression models, one used a Hurdle model, one CUSUM-based methods, one a CART, one a Polynomial distributed lag model and one a negative binomial regression model (Table 6). Seven of the studies were applied to predict RRV outbreaks, one predicted incidence and two predicted cases. Most of the models achieved accuracies or overall agreements of 75.0% or higher.

## Discussion

This systematic review provides a complete analysis of predictive models and exposures for predicting RRV incidence. In contrast to existing reviews which described the climatic, environmental and social factors incorporated in models, this review focuses on the modelling approaches and model performance. Most predictive models used generalised linear models and time series methods, but few studies presented model performance statistics. Many exposures have been included in these models; most of them are in one or two studies only. Rainfall and temperature are the most common exposures, and within the ranges studied, the association with RRV incidence is positive for both exposures in general. Mosquito abundance has a positive effect on RRV as expected.

Data quality was assessed in few studies. This is perhaps because data were collected from government or other public data repositories; consequently, data quality is implicitly considered to be good or the quality is difficult to assess. Some models (e.g. spatial analyses) are unable to predict disease frequency and consequently model evaluation or validation approaches cannot be applied.

**Table 2. Characteristics of the models predicting RRV incidence rates.**

| Study (first author, year) | Research area and period | Model* | Covariates (significant predictors in bold)** |
|---|---|---|---|
| Ryan, 1999 | Maroochy Shire, 1991–1996 | LIRM | **Mosq** |
| Muhar, 2000 | Brisbane, 1991–1996 | LIRM | **Vegetation** types obtained by principal component factor analysis |
| Tong, 2001 | Cairns, 1985–1996 | ARIMA | **Rain**, Temp, **Humd**, Tide |
| Done, 2002 | Queensland, 1991–1997 | Second-order auto-regression model | Stratospheric Quasi-Biennial Oscillation index, the annual, semi-annual and quasi-biennial cycles **(significant predictors not presented)** |
| Tong, 2002(1) | Queensland, 1985–1996 | ARIMA | **Rain**, Temp, **Tide**, adjustment for auto-regression, moving average, seasonality, and year |
| Tong, 2002(2) | Queensland, 1985–1996 | PTSRM | **Rain**, **Temp**, **Humd**, **Tide**, adjustment for seasonality, case notification time and human population (offset variable) |
| Hu, 2004 | Brisbane, 1985–2001 | SARIMA | **Rain**, Temp, Humd, Tide, **auto-regression**, **moving average**, **seasonal auto-regression**, **seasonal moving average** |
| Tong, 2004 | Townsville, 1985–1996 | SARIMA | **Rain**, Temp, Humd, **Tide** |
| Gatton, 2004 | Queensland, 1991–2001 | Spatial analysis | **Spatial autocorrelation** |
| Tong, 2005 | Brisbane, 1998–2001 | PTSRM | **Mosq**, **Rain**, **Tide**, SOI, adjustment for the autocorrelation and seasonality |
| Hu, 2006(1) | Brisbane, 1998–2001 | PTSRM and CART | **Mosq**, adjustment for overdispersion, Max Temp, autocorrelation, and seasonality |
| Hu, 2006(2) | Brisbane, 1998–2001 | PDL and SARIMA | **Mosq**, Rain, Tide, SOI, **seasonal auto-regression** for SARIMA model, adjustment for seasonality and autocorrelation |
| Ryan, 2006 | Redland Shire, 1991–2001 | Spatial analysis | Mosq **(significant predictors not presented)** |
| Hu, 2007 | Brisbane, 2001 | NBRM | **SOI**, human population, overseas visitors, indigenous population, labor workers, **educational level**, family income, **vegetation**, seasonality |
| Jardine, 2008 | Southwestern Australia, 1988–2006 | NBRM and Besag, York, and Mollié (BYM) model | Land salinity, waterlogging, human population (offset variable) |
| Bi, 2009 | The Riverland region of South Australia, 1992–2004 | Poisson regression model | **Rain**, **Temp**, **Humd**, **SOI**, **River Murray flow**, **historical RRV cases**, adjustment for autocorrelation, seasonality and lagged effect |
| Hu, 2010 (1) | Queensland, 1999–2001 | Bayesian spatio-temporal conditional auto-regressive model | **Rain**, **Temp**, SEIFA, **spatial variation**, **LGA-specific temporal trends**, a seasonally oscillating temporal random effect, human population (offset variable) |
| Hu, 2010 (2) | Brisbane, 1998–2001 | CART | **Mosq** |
| Pelecanos, 2011 | Queensland, 1995–2007 | Spatial and temporal analysis | **Spatial autocorrelation** |
| Vally, 2012 | Three regions in Western Australia, 1995–1996 | Poisson linear regression model | **Distance from the waterway**, human population (offset variable) |
| Yu, 2014 | Queensland, 2001–2011 | Spatial and temporal analysis | **Spatial autocorrelation** |
| Koolhof, 2017 | Five sites in Western Australia, 1991–2014 | LIRM | **Rain**, **Temp**, **Tide** |
| Stratton, 2017 | Australia, 1993–2015 | Poisson linear regression model | Rain, Temp or TSI, trend, seasonality, **intra-annual periodicity**, inter-annual periodicity, time-lag |

\* LIRM = Linear Regression Model; ARIMA = Auto-Regressive Integrated Moving Average model; SARIMA = Seasonal Auto-Regressive Integrated Moving Average model; PTSRM = Poisson Time Series Regression Model; CART = Classification and Regression Tree; PDL = Polynomial Distributed Lag model; NBRM = Negative Binomial Regression Model.

\*\*Mosq = Mosquito abundance or Mosquito related data; Rain = Rainfall; Temp = Temperature; Tide = Tide Height or High Tidal level; Humd = Humidity or Relative Humidity; SOI = Southern Oscillation Index; SEIFA = Socio-Economic Index for Areas; TSI = Temperature Suitability Index.

This systematic review identified more than 60 exposures. Climate and weather influence mosquito breeding and behaviour of hosts, and therefore change the prevalence of the disease in a complex way [64]. The lag periods for climatic exposures differ for different parts of the transmission system. Weather can accelerate or decelerate mosquito breeding over a period of several days to weeks [11, 34, 65, 66], while humans may adjust their behaviour immediately in response to weather changes, and host population structure and consequently seroprevalence

**Table 3. Characteristics of the models predicting RRV outbreaks.**

| Study (first author, year) | Research area and period | Model* | Covariates (significant predictors in bold)** |
|---|---|---|---|
| Maelzer, 1999 | New South Wales and Victoria, 1928–1998 | LORM | **SOI** |
| Woodruff, 2002 | Two regions in southeastern Australia, 1991–1999 | LORM | **Rain**, **Temp**, **Humd**, Evap, **VP**, SOI, **SST**, adjustment for irrigation method |
| Gatton, 2004 | Queensland, 1991–2001 | Temporal analysis | **Seasonality** |
| Gatton, 2005 | Queensland, 1991–2001 | LORM | Rain, Temp |
| Woodruff, 2006 | Southwest Western Australia, 1991–1999 | LORM | **Mosq**, **Rain**, Temp, **Tide**, Evap, SOI, SST, VP |
| Watkins, 2008 | Western Australia, 1991–2004 | EARS and NBC | Historical RRV cases (**covariates not quantified**) |
| Pelecanos, 2010 | Queensland, 1991–2007 | EARS, NBC, HLM, POD and temporal analysis | Historical RRV cases (**covariates not quantified**) |
| Sparks, 2010 | New South Wales, 1995–2006 | An adaptive CUSUM plan | Historical RRV cases, day of the week, school holiday, seasonal and transitional influences (**covariates not quantified**) |
| Jacups, 2011 | Northern Territory, 1991–2007 | LORM | **Rain**, **Temp**, **Humd**, **Tide** (only applied for coastal areas) |
| Koolhof, 2017 | Five sites in Western Australia, 1991–2014 | Hurdle model | **Rain**, **Temp**, **Tide** |
| Walsh, 2018 | Australia, 1996–2016 | Maxent model | Rain, Temp, proximity to each surface water type, **proximity to controlled water reservoirs**, **water-soil balance**, **hydrological flow accumulation**, **altitude**, MGVF, human migration, **ecological niches of wildlife hosts** |
| Tall, 2019 | Inland New South Wales, 1991–2013 | GEE | **Flood events** |

* LORM = Logistic Regression Model; EARS = Early Aberration Reporting System C1, C2 and C3 algorithms; CUSUM = Cumulative Sum; NBC = Negative Binomial Cusum method; HLM = Historical Limits Method; POD = Poisson Outbreak Detection method; GEE = Generalised Estimating Equations.

**Mosq = Mosquito abundance or Mosquito related data; Rain = Rainfall; Temp = Temperature; Tide = Tide Height or High Tidal level; Humd = Humidity or Relative Humidity; Evap = Evaporation; VP = Vapor Pressure; SOI = Southern Oscillation Index; SST = Sea Surface Temperature; MGVF = Maximum Green Vegetation Fraction.

may be affected by climate after a few years [10, 67, 68]. This phenomenon also explains why the same exposure can influence RRV incidence both positively and negatively at different lag times. Interactions between climatic exposures further complicate the analysis [65].

Data on vectors and reservoir host species, abundance and competence are crucial for forecasting RRV incidence [60, 69, 70]. The importance of vectors and reservoir hosts differs between species because of behavioural and ecological variation [71, 72]. The feeding and breeding of mosquito species are affected by host availability and abundance [73, 74]. Across urban, inland and coastal regions of Australia, vector and host species driving RRV transmission are diverse and variable [2]. Because of the wide variety of non-human reservoir hosts, it is extremely difficult to ascertain the complex relationships among hosts, vectors, and disease incidence. Epidemiological analyses and host ecology studies including serosurveys are important methods of detecting and describing these relationships. However, vector and reservoir host data with sufficient details and completeness to be useful for prediction are rarely available, impairing the quality of models.

Surface water sources, river flow, vegetation and remoteness, which were included only in a few studies, are promising data sources and should be explored further. Surface water and vegetation provide a favourable environment for mosquito breeding and are important for modelling [75, 76]. These exposures are increasingly incorporated in recent models [53, 61]. Inclusion of incidence terms from past weeks is also widely used in public health surveillance,

**Table 4. Characteristics of the models predicting RRV cases.**

| Study (first author, year) | Research area and period | Model* | Covariates (significant predictors in bold)** |
|---|---|---|---|
| Jacups, 2008 | Darwin, 1991–2006 | Poisson linear regression model | **Mosq**, **Rain**, **Temp**, Humd, Tide |
| Barton, 2009 | The Gippsland Lakes region of eastern Victoria, 1991–2001 | LIRM | **Mosq**, **Rain**, **Temp** |
| Williams, 2009 | The River Murray Valley of South Australia, 1999–2006 | LIRM | **Mosq**, **Rain**, **Temp**, **river height**, human population |
| Werner, 2012 | Southeastern Tasmania, 1993–2009 | NBRM | **Rain**, **Temp**, Tide |
| Ng, 2014 | Four regions in New South Wales, 1991–2004 | PDL | **Mosq**, **Rain**, **Temp**, **Humd**, **Tide**, **Evap**, **SST**, **NDVI**, **water sources**, distance to coast, elevation, **ARIA**, **macropod population**, human population (offset variable) |
| Rohart, 2016 | Australia, 2009–2013 | High-dimensional LIRM | Google Trends data (**covariates not quantified**) |
| Cutcher, 2017 | The Mildura Local Government Area of Victoria, 2000–2015 | NBRM | **Mosq**, **Rain**, Temp, Humd, **VP**, **SOI**, La Niña events, **SST**, sea level, **river flow** |
| Flies, 2018 | South Australia, 1992–2012 | Generalised additive model | Mosq, Rain, **Temp**, Humd, **distance to coast**, **distance to Murray River**, NDVI, **elevation**, **IRSD**, Caravan parks, **global human settlement "urban-ness" score**, **non-human reservoirs**, **expected RRV cases** |
| Walker, 2018 | The Peel region of southwest Western Australia, 2003–2014 | NBRM | **Mosq**, **presence of RRV isolates**, **season** |
| Koolhof, 2019 | The eleven Local Government Areas in Victoria, 2005–2018 | NBRM | **Rain**, **Temp**, **Humd**, **Tide**, **VP**, **sea level pressure**, **Evap**, SOI, **SST**, **river flow** |

* LORM = Logistic Regression Model; LIRM = Linear Regression Model; PDL = Polynomial Distributed Lag model; NBRM = Negative Binomial Regression Model.

**Mosq = Mosquito abundance or Mosquito related data; Rain = Rainfall; Temp = Temperature; Tide = Tide Height or High Tidal level; Humd = Humidity or Relative Humidity; Evap = Evaporation; VP = Vapor Pressure; SST = Sea Surface Temperature; SOI = Southern Oscillation Index; NDVI = Normalized Difference Vegetation Index; ARIA = Accessibility/Remoteness Index of Australia; IRSD = Index of Relative Socio-economic Disadvantage.

e.g. the Early Aberration Reporting System, which offers aberration detection methods by analysing recent surveillance data [77].

The time-lag effects of RRV activity are generated not only by climatic factors but also by mosquito abundance, host populations and some geographical elements such as river flow and flooding [9, 15, 52, 56, 63, 78]. The time lags are also influenced by the species diversity and abundance of mosquitoes in the research area. For instance, the freshwater-breeding *Culex annulirostris* is affected by rainfall and riverine flooding at freshwater habitats, while the estuarine-breeding *Aedes vigilax* is associated with estuarine wetlands shaped by tidal flooding and rainfall [2]. Thus, analysing temporal data is helpful to identify the temporal variation in these associations with RRV incidence. Moreover, the host population, mosquito breeding and people's lifestyle vary spatially. Data on the geographical difference and temporal trends of related exposures can be valuable for RRV prediction.

Our systematic analysis showed that linear models and time-series approaches are the two main analytical methods used to predict RRV disease. Linear regressions are simple to manipulate and explain, while time-series models are appropriate for considering autocorrelation and seasonal fluctuations. Both approaches have been widely applied in dealing with infectious diseases [79, 80]. Their pros and cons are described in some articles [81–83]. Models with good predictive performance perform well at predicting outcomes for out-of-sample data [84]. Usually cross-validation is used to assess model performance in retrospective studies, and 25% of available data for validation is recommended [84]. Some statistics can be derived, such as accuracy, specificity, sensitivity, mean-squared error, mean absolute error or root mean-squared error, for evaluation [85]. Head-to-head comparisons of models using common

**Table 5. Top 15 significant exposures applied, and the directions of their associations with RRV.**

| Exposures* | $N_{sa}$ / $N_a$** | $N_{sm}$ / $N_m$** | Association with RRV (Positive / Negative) *** |
|---|---|---|---|
| Rainfall | 20 / 25 | 45 / 69 | 47 / 14 |
| Temperature or Temperature Suitability Index | 14 / 23 | 35 / 66 | 31 / 15 |
| Tidal height or high tidal level | 10 / 15 | 19 / 45 | 17 / 6 |
| Mosquitoes, with top three: *Culex annulirostris*, *Aedes camptorhynchus* and *Culex australicus* | 12 / 14 | 23 / 42 | 43 / 2 |
| Humidity or relative humidity | 7 / 12 | 12 / 37 | 5 / 10 |
| Southern Oscillation Index | 4 / 9 | 6 / 15 | 5 / 1 |
| Sea Surface Temperature | 4 / 5 | 6 / 22 | 4 / 6 |
| River flow or river height | 5 / 5 | 5 / 19 | 5 / 2 |
| Vapor pressure | 3 / 4 | 5 / 18 | 5 / 0 |
| Distance to each surface water type | 3 / 4 | 3 / 9 | 0 / 4 |
| Evaporation | 2 / 4 | 7 / 21 | 8 / 1 |
| Seasonality / Season | 2 / 4 | 7 / 9 | 7 / 0 |
| Historical RRV cases | 1 / 4 | 2 / 24 | Not quantified |
| Non-human reservoir hosts: Grey kangaroos, birds, or mammals | 3 / 3 | 7 / 8 | 7 / 3 |
| Elevation / Altitude | 2 / 3 | 2 / 8 | 2 / 0 |

* Variables used as offsets or used for adjustment were not included, interactions of the variables were not included.

** $N_{sa}$ is the number of articles that have significant exposures; $N_a$ is the number of articles that used the exposures; $N_{sm}$ is the number of models that have significant exposures; $N_m$ is the number of models that used the exposures. Numbers are summarised in categories of mosquitoes and non-human reservoir hosts, which indicates one or more species are applied in each article or each model.

*** The same exposure can be applied several times with different time periods or time-lags in the same model. The associations of significant exposures and RRV infections were not quantified in some papers.

datasets are suggested for model assessment [86]. Robustness of the models need to be tested under various settings [86]. The best modelling approach for RRV prediction is currently unclear. Therefore, the performances of RRV predictive models are needed in order to compare them and select the best one for a given setting.

This is the first systematic review focusing on modelling approaches for predicting RRV disease. This is also the first review that lists statistical methods, significant exposures and the modelling performance of selected studies. Only studies conducted in Australia and published in English were included. We did not search grey literature. We were unable to evaluate publication bias, however two of the included studies were published without significant results. Although we have summarised broad findings in relation to exposures, we have not conducted meta-analysis. Insufficient data were available to assess performance of most models. Therefore, we were not able to strictly compare models and establish an appropriate level of confidence in their performance. The descriptions of predictive models included in this review are based on current publications, so, these data need to be interpreted with caution. The information we have presented is dependent on the limitations of the included studies. A particular issue in RRV research is the non-equivalence between routinely collected surveillance data and RRV incidence. There are also significant limitations for exposure data, brought about by site and number of weather stations, incompleteness of macropod data, variability in mosquito enumeration due to characteristics of particular trap types, and other issues.

**Table 6. Model performance.**

| Study (author, year) | Model* | Prediction performance |
|---|---|---|
| Maeizer, 1999 | LORM | Positive predictive value = 88%, negative predictive value = 91%. |
| Woodruff, 2002 | LORM | Early warning models: accuracies were 64% - 100%. Late warning models: accuracies were 63% - 100%. |
| Gatton, 2005 | LORM | Across regions, accuracies were 88%-98%, sensitivities were 0.53–0.83, and specificities were 0.94–1.00. |
| Hu, 2006(1) | CART | Overall agreement = 76%, Sens/Spec = 0.61/0.80. |
| Woodruff, 2006 | LORM | Early warming model: Sens/Spec = 0.90/0.88. Late warming model: 0.85/0.98. |
| Pelecanos, 2010 | EARS, NBC, HLM, POD and temporal analysis | True positives for four regions were 40% - 89%, 13% - 75%, 0% - 100%, 19% - 100%. |
| Jacups, 2011 | LORM | Sensitivity/Specificity crossover were 75.8% - 88.5%. |
| Ng, 2014 | PDL | Across regions, accuracies were 68.7% - 84.7%. |
| Koolhof, 2017 | Hurdle model | Across regions, sensitivities were 0–1, specificities were 0.18–1. |
| Koolhof, 2019 | NBRM | Pearson's correlation coefficients of predicted and observed notifications $\geq 0.6$ in 5 locations (5/11, 45.5%). |

* LORM = Logistic Regression Model; CART = Classification and Regression Tree; EARS = Early Aberration Reporting System C1, C2 and C3 algorithms; NBC = Negative Binomial Cusum method; HLM = Historical Limits Method; POD = Poisson Outbreak Detection method; PDL = Polynomial Distributed Lag model; NBRM = Negative Binomial Regression Model.

Given the complex transmission cycle of the virus, exposures and RRV incidence would not be expected to have a simple linear relationship. Non-linear models such as generalised additive mixed models and machine learning approaches are more likely to provide a more sophisticated representation of the transmission system than linear regression [87–89]. Analytical methods that encompass climate, environmental exposures, socio-economic factors and spatio-temporal aspects for forecasting RRV incidence are also worthy of consideration. For example, Bayesian spatio-temporal modelling by Hu (2010) [45] considered the spatial effects, temporal trends, climatic exposures and an interaction term for climate exposures. Region-specific models are ideal, due to spatial variation in transmission [53]. The complex ecology and the environmental variation in Australia make it challenging to design models with universal applicability that are useful for public health programs. However, there is benefit in assessing the performance of these models, as we have done in this review, to determine usefulness, even if this means rejecting some approaches. Our work will continue with the development of RRV models for Queensland using innovative modelling approaches and then assessing their predictive performance.

Our systematic review provides an analysis of epidemiological models for predicting RRV disease using notification data in Australia. Current modelling approaches are valuable in improving understanding of RRV transmission and in predicting outbreaks. However, model performance assessments are notably lacking. Nonetheless, the summary of significant exposures provided in our systematic review offers suggestions for future modelling. Predictive models are definitely useful tools for understanding transmission and predicting outbreaks of RRV. Better data availability, combined with new modelling approaches and performance assessment may improve the accuracy of forecasting. More detailed information, like daily or weekly data on RRV cases and climatic exposures at a smaller spatial scale will improve model prediction performance [53]. RRV ecology research that provides data on the abundance or

spatio-temporal distribution of the mosquitoes and non-human reservoir hosts is beneficial for modelling the transmission cycle and forecasting disease incidence.

## Supporting information

**S1 Table. PRISMA 2009 checklist.**
(DOC)

**S2 Table. Quality assessment criteria.**
(DOCX)

**S3 Table. Quality assessment scoring details.**
(DOCX)

**S4 Table. Details of the exposures included in models.**
(DOCX)

**S1 Text. Protocol of systematic review.**
(DOCX)

**S2 Text. Search terms.**
(DOCX)

## Acknowledgments

We are grateful to Mrs Wai Wai Lui, the Senior Librarian of Mater McAuley Library, University of Queensland, who supported us greatly in the literature search. Australian governments fund the Australian Red Cross Lifeblood to provide blood, blood products and services to the Australian community.

## Author Contributions

**Conceptualization:** Wei Qian, Elvina Viennet, Kathryn Glass, David Harley.

**Data curation:** Wei Qian, Elvina Viennet, Kathryn Glass, David Harley.

**Formal analysis:** Wei Qian.

**Funding acquisition:** Wei Qian, Elvina Viennet, David Harley.

**Supervision:** Elvina Viennet, Kathryn Glass, David Harley.

**Visualization:** Wei Qian.

**Writing – original draft:** Wei Qian.

**Writing – review & editing:** Wei Qian, Elvina Viennet, Kathryn Glass, David Harley.

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
