## [Decision Letter · Decision Letter 0]

9 Jun 2020

Dear Associate Professor Harley,

Thank you very much for submitting your manuscript "Epidemiological models for predicting Ross River virus in Australia: a systematic review" for consideration at PLOS Neglected Tropical Diseases. As with all papers reviewed by the journal, your manuscript was reviewed by members of the editorial board and by several independent reviewers. The reviewers appreciated the attention to an important topic. Based on the reviews, we are likely to accept this manuscript for publication, providing that you modify the manuscript according to the review recommendations. 

Since there was no third reviewer I also evaluated the manuscript. I thought the authors did a very nice job of evaluating the literature. I have a few specific comments. 1) Did one author review all possible articles or did multiple authors make the decisions about inclusion and assessment of the outcomes? And if multiple authors reviewed the manuscripts how was care taken to ensure they assessed the quality/inclusion in similar ways? 2) The authors comment that meta-analysis was not possible due to a lack of similar models and model evaluations in the literature. However they do present some nice tables showing which factors predicted incidence/outbreaks/cases with Table 5 trying to summarize the top 15 predictors. In the discussion can you comment on those top 15 and expand. Some are discussed well (vectors, enviromental), but some are less well discussed (hosts). 3) In table 5 when you say "positive or negative" association with RRV can you clarify what the reference variable would be for each of the categories? i.e. is a positive association with non human reservoirs that more reservoirs = more incidence? Etc. Other than those comments I thought the manuscript was well written and look forward to the revisions.

Sincerely,

Brianna R Beechler, Ph.D., DVM

Guest Editor

Amy Morrison

Deputy Editor

The reviewers recommended only minor revisions. Since there was no third reviewer I also evaluated the manuscript. I thought the authors did a very nice job of evaluating the literature. I have a few specific comments. 1) Did one author review all possible articles or did multiple authors make the decisions about inclusion and assessment of the outcomes? And if multiple authors reviewed the manuscripts how was care taken to ensure they assessed the quality/inclusion in similar ways? 2) The authors comment that meta-analysis was not possible due to a lack of similar models and model evaluations in the literature. However they do present some nice tables showing which factors predicted incidence/outbreaks/cases with Table 5 trying to summarize the top 15 predictors. In the discussion can you comment on those top 15 and expand. Some are discussed well (vectors, enviromental), but some are less well discussed (hosts). 3) In table 5 when you say "positive or negative" association with RRV can you clarify what the reference variable would be for each of the categories? i.e. is a positive association with non human reservoirs that more reservoirs = more incidence? Etc. Other than those comments I thought the manuscript was well written and look forward to the revisions.

Reviewer's Responses to Questions

**Key Review Criteria Required for Acceptance?**

**Methods**

-Are the objectives of the study clearly articulated with a clear testable hypothesis stated?

-Is the study design appropriate to address the stated objectives?

-Is the population clearly described and appropriate for the hypothesis being tested?

-Is the sample size sufficient to ensure adequate power to address the hypothesis being tested?

-Were correct statistical analysis used to support conclusions?

-Are there concerns about ethical or regulatory requirements being met?

Reviewer #1: The methods are clearly defined as well as the aim of the study. The study design is appropriate and with a sufficient number of studies the meet the eligible criteria defined in the methods section. There is only the need to clarify the number of researches that participated in the searches done in the different databases.

Reviewer #2: I have no concerns regarding methodology of this manuscript. All matters listed above appear to have been addressed satisfactorily to my knowledge.

**Results**

-Does the analysis presented match the analysis plan?

-Are the results clearly and completely presented?

-Are the figures (Tables, Images) of sufficient quality for clarity?

Reviewer #1: The results are clearly presented, with a clear flow chart of the selection process and with enough detail on the quality grading and process.

Reviewer #2: I have no concerns regarding the results presented in this manuscript. All matters listed above appear to have been addressed satisfactorily.

**Conclusions**

-Are the conclusions supported by the data presented?

-Are the limitations of analysis clearly described?

-Do the authors discuss how these data can be helpful to advance our understanding of the topic under study?

-Is public health relevance addressed?

Reviewer #1: The conclusion is clearly stated and supported by the findings. The benefits in predicting RRV disease through the use models is stablished as well as the gaps in the use and performance assessment of the models. However, the study lacks an identification of limitations of the study and how these were addressed.

Reviewer #2: I appreciate that the authors made some comments at the end of their discussion on general usefulness of models and data that would be provide greater benefit for inclusion in future. There is a perennial debate among many mosquito and public health professionals on the utility of predictive models or mosquito-borne disease. It is often claimed that deficiencies in accuracy, resulting from the ecological complexity of RRV transmission cycles, mean that they are of only marginal use to local authorities in practical/operational responses. I’d like to see some additional comments from authors about the translation of this work to inform this decision making process. Are the authors recommended a particular type of model (or combination of models) as being more useful in certain circumstances? In addition, can more specific suggestions be made about the application of these models to certain geographic areas/environments or, alternatively, future modelling approaches applicable to these settings.

The authors should also include some comments on the limitations of this work. In particular is the inconsistencies in the scope of the existing literature. While I understand that the authors are only working with published literature, there will remain some biases in that published literature that may have implications for the adaptation of results elsewhere in the country. This is especially the case where regions have not been a focus of previous modelling attempts or where some regions (e.g. NT or WA) have relatively greater focus. 

Some minor comments.

Minor comments:

Line 61. Add “tides” to list of environmental factors 

Line 242-243. It may be worth making a note that these differences in lag between environmental/climatic factors will be influenced by differences in vector abundance/diversity within regions studied. Here, or later starting lines 265, it may be worth noting the distinct differences in key mosquitoes associated with freshwater (Culex annulirostris) and estuarine environments (Aedes vigilax).

**Editorial and Data Presentation Modifications?**

Reviewer #1: Only minor revisions are needed in the publication

Reviewer #2: The manuscript is well written and presented and I have no major comments on the methodology or analysis. Reviews of this nature are useful for those looking to investigate strategies to develop predictive models of mosquito-borne disease to assist health authorities manage risk.

**Summary and General Comments**

Reviewer #1: The study is well developed and provides important information regarding the use of different models to predict RRV disease, therefore the study can provide useful resources to already stablished surveillance systems.

Reviewer #2: See comments above regarding conclusions.

PLOS authors have the option to publish the peer review history of their article (what does this mean?). If published, this will include your full peer review and any attached files.

Reviewer #1: No

Reviewer #2: No
---

## [Decision Letter · Decision Letter 1]

20 Jul 2020

Dear Associate Professor Harley,

We are pleased to inform you that your manuscript 'Epidemiological models for predicting Ross River virus in Australia: a systematic review' has been provisionally accepted for publication in PLOS Neglected Tropical Diseases.

Best regards,

Brianna R Beechler, Ph.D., DVM

Guest Editor

Amy Morrison

Deputy Editor

Thank you for completely addressing all comments provided by the reviewers.

Reviewer's Responses to Questions

**Key Review Criteria Required for Acceptance?**

**Methods**

-Are the objectives of the study clearly articulated with a clear testable hypothesis stated?

-Is the study design appropriate to address the stated objectives?

-Is the population clearly described and appropriate for the hypothesis being tested?

-Is the sample size sufficient to ensure adequate power to address the hypothesis being tested?

-Were correct statistical analysis used to support conclusions?

-Are there concerns about ethical or regulatory requirements being met?

Reviewer #1: The observations from the methods section were correctly addressed

Reviewer #2: All methods in the revised manuscript satisfactory.

**Results**

-Does the analysis presented match the analysis plan?

-Are the results clearly and completely presented?

-Are the figures (Tables, Images) of sufficient quality for clarity?

Reviewer #1: There were no observations from the results section

Reviewer #2: All results presented satisfactory.

**Conclusions**

-Are the conclusions supported by the data presented?

-Are the limitations of analysis clearly described?

-Do the authors discuss how these data can be helpful to advance our understanding of the topic under study?

-Is public health relevance addressed?

Reviewer #1: The observations regarding the imitations were correctly addressed

Reviewer #2: All conclusions satisfactory and I acknowledge that the authors have addressed my suggestions for inclusion of additional discussion.

**Editorial and Data Presentation Modifications?**

Reviewer #1: Accept

Reviewer #2: N/A

**Summary and General Comments**

Reviewer #1: There are no new comments

Reviewer #2: A good paper and makes a useful contribution to understand the risks of mosquito-borne disease in Australia.

PLOS authors have the option to publish the peer review history of their article (what does this mean?). If published, this will include your full peer review and any attached files.

Reviewer #1: No

Reviewer #2: No

---

## [Editor Report · Acceptance letter]

9 Sep 2020

Dear Associate Professor Harley,

We are delighted to inform you that your manuscript, "Epidemiological models for predicting Ross River virus in Australia: a systematic review," has been formally accepted for publication in PLOS Neglected Tropical Diseases.

Best regards,

Shaden Kamhawi

co-Editor-in-Chief

Paul Brindley

co-Editor-in-Chief
